# Fully-gapped superconductivity with rotational symmetry breaking in pressurized kagome metal CsV$_3$Sb$_5$

X. Y. Feng[1,2], Z. Zhao[1,2], J. Luo[1], Y. Z. Zhou[1,2], J. Yang[1], A. F. Fang[3,4], H. T. Yang[1,2], H.-J. Gao[1,2], R. Zhou[1,2] ✉ & Guo-qing Zheng[5]

The discovery of the kagome metal CsV$_3$Sb$_5$ has generated significant interest in its complex physical properties, particularly its superconducting behavior under different pressures, though its nature remains debated. Here, we performed low-temperature, high-pressure $^{121/123}$Sb nuclear quadrupole resonance (NQR) measurements to explore the superconducting pairing symmetry in CsV$_3$Sb$_5$. At ambient pressure, we found that the spin-lattice relaxation rate $1/T_1$ exhibits a kink at $T \sim 0.4\,T_c$ within the superconducting state and follows a $T^3$ variation as temperature further decreases. This suggests the presence of two superconducting gaps with line nodes in the smaller one. As pressure increases beyond $P_c \sim 1.85$ GPa, where the charge-density wave phase is completely suppressed, $1/T_1$ shows no Hebel-Slichter peak just below $T_c$, and decreases rapidly, even faster than $T^5$, indicating that the gap is fully opened for pressures above $P_c$. In this high pressure region, the angular dependence of the in-plane upper critical magnetic field $H_{c2}$ breaks the $C_6$ rotational symmetry. We propose the $s + id$ pairing at $P > P_c$ which explains both the $1/T_1$ and $H_{c2}$ behaviors. Our findings indicate that CsV$_3$Sb$_5$ is an unconventional superconductor and its superconducting state is even more exotic at high pressures.

The kagome lattice, formed by a network of corner-sharing triangles, features Dirac fermions, flat bands, and van Hove singularities in its electronic structure. This makes it an ideal system for exploring geometric frustration, strongly correlated electronic states, and topological quantum phenomena[1–3]. The Dirac cones support massless charge carriers, while flat bands result in highly localized electrons through destructive interference[4]. This electron localization amplifies electron-electron interactions, potentially fostering exotic phenomena such as quantum spin liquid states[1,5,6], fractional quantum Hall states[7,8], and unconventional superconductivity[9,10]. While theoretical predictions suggest the unconventional pairing in kagome-lattice superconductors, most discovered materials with this structure exhibit conventional superconducting behaviors. For instance, CeRu$_2$, one of

the earliest superconductors with a kagome lattice[11,12], shows an $s$-wave gap symmetry[13]. Recently, the Copper(II)-based coordination polymer Cu-BHT[14,15] emerged as a potential unconventional superconductor, exhibiting nonexponential temperature-dependent superfluid density, although further experiments are required to confirm its superconducting gap structure.

The $A$V$_3$Sb$_5$ ($A$ = K, Cs, Rb) family, another kagome-lattice superconductor group, has recently attracted interests due to its unique electronic properties[16–22]. At ambient pressure, CsV$_3$Sb$_5$ undergoes a charge density wave (CDW) transition at $T_{CDW} = 94$ K, followed by a superconducting transition at $T_c = 2.5$ K[23]. This CDW state is associated with unusual phenomena such as chirality[24–26], nematicity[27], and time-reversal symmetry breaking (TRSB)[28]. Hydrostatic pressure gradually

[1]Institute of Physics, Chinese Academy of Sciences, and Beijing National Laboratory for Condensed Matter Physics, Beijing 100190, China. [2]School of Physical Sciences, University of Chinese Academy of Sciences, Beijing 100190, China. [3]School of Physics and Astronomy, Beijing Normal University, Beijing 100875, China. [4]Key Laboratory of Multiscale Spin Physics, Ministry of Education, Beijing Normal University, Beijing 100875, China. [5]Department of Physics, Okayama University, Okayama 700-8530, Japan. ✉e-mail: rzhou@iphy.ac.cn

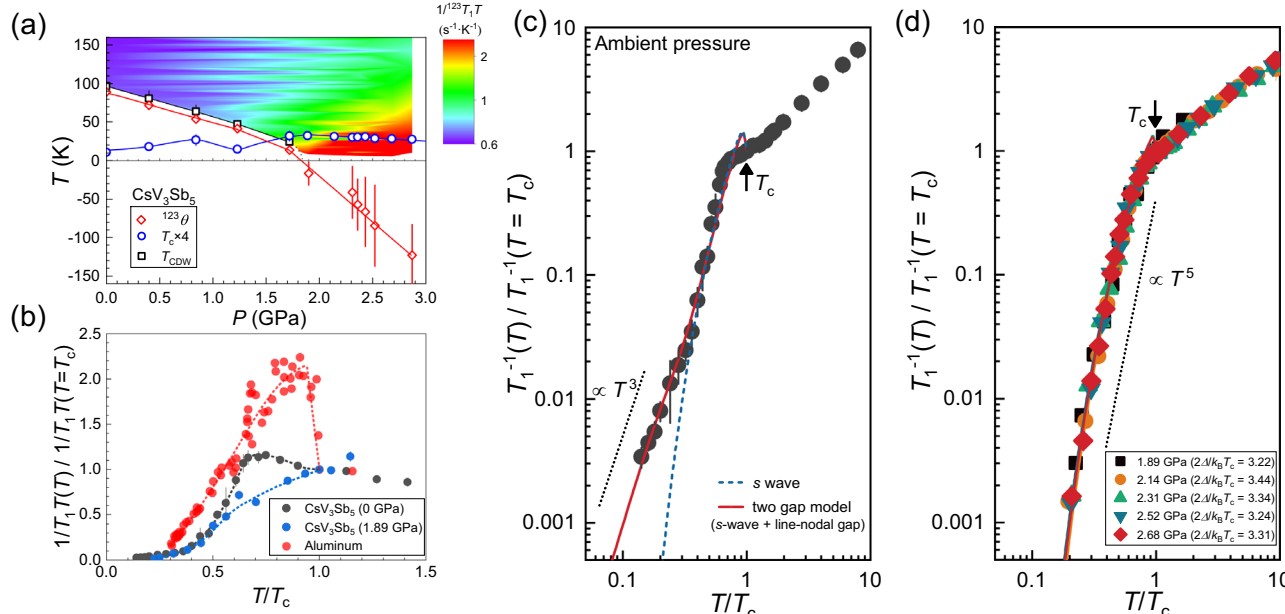

**Fig. 1 | The phase diagram of CsV₃Sb₅ and temperature-dependent 1/T₁ in the superconducting state. a** The black squares represent $T_{CDW}$ values derived from our previous NQR study[31], while blue circles indicate $T_c \times 4$, obtained from ac susceptibility measurements[47]. Red diamonds mark $^{123}\theta$, extracted from Curie-Weiss fitting of the $^{123}$Sb2 NQR linewidth (see Supplementary Fig. 4(a))[47]. Color variations above $T_{CDW}$ reflect the evolution of $1/T_1T$ of $^{123}$Sb2 (see Supplementary Fig. 4(b))[47]. Solid and dashed lines are guides for the eye. The error bar for $\theta$ is the s.d. in the fitting of linewidth. The error bar for $T_{CDW}$ reflects the temperature interval used in NQR spectra measurements[31]. The error bar for $T_c$ corresponds to the transition width evaluated from the 10-90% criterion. **b** The black and blue dots represent temperature-dependent $1/T_1T$ at ambient pressure and 1.89 GPa, respectively. The red dots represent temperature-dependent $1/T_1T$ of aluminum[48].

The $1/T_1T$ of aluminum shows a clear Hebel-Slichter peak below $T_c$, while just a small Hebel-Slichter peak below $T_c$ was observed in CsV₃Sb₅. Above $P_c$, the Hebel-Slichter peak is absent below $T_c$(see Supplementary Fig. 5 for $P \geq 2.14$ GPa)[47]. The dashed lines are guides for the eye. **c** Temperature dependence of the normalized spin-lattice relaxation rates $[1/T_1(T)/1/T_1(T_c)]$ at ambient pressure. The blue dashed curve represents the simulation of the $s$-wave model, while the red solid curve represents the simulation of the $s$-wave + line-nodal gap model. **d** Temperature dependence of the normalized spin-lattice relaxation rates $[1/T_1(T)/1/T_1(T_c)]$ at $P > 1.85$ GPa. Solid lines represent the simulations of the fully-gapped model with a single energy gap. The dotted line indicates $T^3$ and $T^5$ behaviors as visual guides in (c) and (d), respectively. The solid arrows indicate $T_c$. The error bar in $T_1$ is the s.d. in fitting the nuclear magnetization recovery curve.

suppresses the CDW transition, which disappears around a critical pressure, $P_c \sim 1.9$ GPa, while $T_c$ forms a double-dome-shaped phase diagram, hinting at the unconventional nature of the superconductivity[29–33]. Some experimental observations indeed suggest unconventional superconducting behaviors in this system. The transport measurements reveal a two-fold rotational symmetry in the superconducting state under ambient pressure[34,35]. Other exotic features of the superconducting state, such as the presence of Majorana zero modes within vortex cores[36], pairing density waves (PDW) and TRSB have also been observed[37–39]. However, the appearance of a Hebel-Slichter peak in NQR measurements[40], temperature-dependent superfluid density from transverse-field muon spin rotation ($\mu$SR) experiments[41,42], and magnetic penetration depth measurements[43] point to an $s$-wave gap. Given its nonmagnetic nature and relatively weak electron correlations, CsV₃Sb₅ is still often proposed to be a phonon-mediated conventional superconductor[44], though this is still under debate. The nature of superconductivity at high pressures is also contested, partly due to a limited number of high-pressure experiments. The $\mu$SR measurements suggest spontaneous TRSB just below $T_c$ and a nodeless superconducting gap when charge order is fully suppressed[45,46]. However, NQR studies find no Hebel-Slichter peak in the spin-lattice relaxation rate $1/T_1$ below $T_c$[30], which appears inconsistent with a fully gapped state. Additional high-pressure measurements are essential to further investigate the unique properties of the superconducting state, which are crucial for understanding the pairing mechanism in CsV₃Sb₅.

In this study, we carried out low-temperature and high-pressure $^{121/123}$Sb-NQR measurements to investigate the pairing symmetry in CsV₃Sb₅. At ambient pressure, a distinct kink was observed in the spin-lattice relaxation rate $1/T_1$ at $T \sim 0.4 T_c$ followed by a $T^3$ behavior down

to the lowest temperatures. This behavior indicates multiple superconducting gaps, with line nodes present in the smaller one. When the pressure exceeds $P_c \sim 1.85$ GPa, where the CDW phase is completely suppressed, the temperature dependence of $1/T_1$ indicates that the superconducting gap fully opens. Most remarkably, the angular dependence of the in-plane upper critical magnetic field, $H_{c2}$, reveals an unexpected two-fold symmetry in the fully-gapped superconducting phase at $P > P_c$. We propose an $s + id$ pairing that simultaneously explains the full gap and rotational-symmetry broken behavior of $H_{c2}$. We will also discuss a possible charge redistribution below $T_c$ in the $P > P_c$ region suggested by an increase in both the NQR frequency and linewidth. Our findings provide new insights into the unconventional superconductivity of kagome metals.

## Results

### Evolution of the superconducting gap symmetry

Figure 1(a) represents the pressure-temperature phase diagram of CsV₃Sb₅[31], incorporating data on the parameter $^{123}\theta$, derived from fitting the NQR linewidth and spin-lattice relaxation rate $1/T_1T$ within the pressure range of 2 to 3 GPa (see Supplementary Fig. 4[47]). The parameter $\theta$ indicates the presence of CDW correlations. As pressure $P$ surpasses the critical value $P_c \sim 1.85$ GPa, a decrease in $\theta$ reflects a reduction in CDW fluctuations. At the same time, an increase in $1/T_1T$ with rising pressure suggests an enhancement in spin fluctuations. In the normal state, $1/T_1T$ increases only slightly with decreasing temperature at $P < P_c$, but is enhanced rapidly when temperature decreases at $P > P_c$. To explore the superconducting gap symmetry under varying pressures, we measured the temperature dependence of $1/T_1$ in the superconducting state. Figure 1(b) displays the temperature

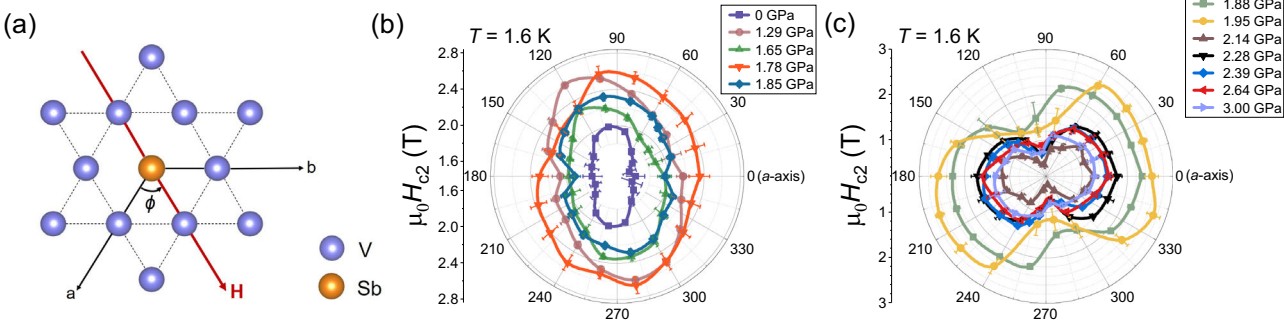

**Fig. 2 | Evidence for persistent two-fold symmetry of $H_{c2}$ at high pressures.**
**a** The illustration depicts the field orientation in the hexagonal plane with respect to the $a$ axis (angle $\phi$), where an angle of $\phi = 0°$ corresponds to $H \| a$ axis. **b**, **c** represent the angular dependence of $H_{c2}$ respectively below and above $P_c$. The data

are obtained from ac susceptibility measurements at a fixed temperature $T = 1.6$ K (see Supplementary Fig. 7)[47]. The error was estimated from uncertainties in the determination of $H_{c2}$ due to scatter in the raw field sweeps.

dependence of $1/T_1T$ for CsV$_3$Sb$_5$. At ambient pressure, we also observe a small Hebel-Slichter coherent peak below $T_c$, consistent with previous NQR studies[30,40]. The presence of this coherence peak typically suggests the $s$-wave symmetry in the superconducting gap. However, its magnitude is notably smaller than that observed in conventional superconductors such as aluminum (see Fig. 1(b))[48]. As temperature decreases, $1/T_1$ drops sharply below $T_c$ but changes to a $T^3$ dependence below $T \sim 0.4$ $T_c$ as shown in Fig. 1(c). The $T^3$ variation is a characteristic behavior of line nodes in the gap function[49,50]. The relaxation rate $1/T_1$ below $T_c$ is expressed as[51]

$$\frac{T_1(T_c)}{T_{1s}} = \frac{2}{k_B T_c} \int \left(1 + \frac{\Delta^2}{E^2}\right) N_s(E)^2 f(E)[1 - f(E)]dE \qquad (1)$$

where $N_s(E) = N_0 E / \sqrt{E^2 - \Delta^2}$ is the density of state (DOS) in the superconducting state, $\Delta$ is the magnitude of the energy gap, $f(E)$ is the Fermi distribution function, and the $(1 + \Delta^2/E^2)$ is the coherence factor. For an $s$-wave gap, the coherence factor and the divergence of the DOS at $E = \Delta$ will lead to a Hebel-Slichter peak just below $T_c$[52,53]. First, we used a single $s$-wave gap model to simulate $1/T_1$, with $\Delta_s = 1.85 k_B T_c$, represented by the blue dashed line in Fig. 1(c). However, we found that the simulation results at low temperatures did not align with the experimental results. Considering that there exists a $T^3$ temperature-dependent relationship at low temperatures, we then used a two gap model, namely an $s$-wave gap $+$ a line-nodal gap($\Delta(\Phi) = \Delta_0 \cos(2\Phi)$), to simulate the $1/T_1$ results.

We assume that the total superconducting DOS is contributed from two gaps as $N_{\text{tot}} = \alpha \cdot N_s + (1 - \alpha) \cdot N_L$, where $N_s$ and $N_L$ are the superconducting DOS from the $s$-wave and line-nodal gap, respectively. Using such two-gap model allowed us to successfully fit the data over the entire temperature range shown in Fig. 1(c), yielding $\Delta_s = 2.0 k_B T_c$ and $\Delta_L = 0.63 k_B T_c$ with a ratio of DOS on two bands ($N_s$: $N_L = 0.9$: 0.1). These results suggest that CsV$_3$Sb$_5$ features a small superconducting gap with line nodes. Nevertheless, both the relative weight and the gap size of the fully-opened gap are found to be much larger than this nodal gap. We also note that theoretical calculations on the kagome lattice have indicated the possibility of unconventional pairing states with a sign-changing gap structure, particularly $d$-wave gap, which may manifest as a small Hebel-Slichter peak in the temperature-dependent $1/T_1$ due to the destructive sublattice interference effects[54]. It is probable that these factors result in a small coherence peak in $1/T_1$ as shown in Fig. 1(c). Given that both the size and the weight of the nodal gap are extremely small compared to the fully opened gap, the signature of this gap can only be observed at very low temperatures, which is indeed in agreement with the observation of a residual DOS in scanning tunneling spectroscopy[37,38]. This is

precisely why it was challenging to detect previously. In any case, our results show that CsV$_3$Sb$_5$ is already an unconventional superconductor at ambient pressure. The small component of the line-nodal gap is consistent with the temperature dependence of $1/T_1T$ in the normal state which points to weak spin fluctuations (see Fig. 1(a)).

Next, we examine the $1/T_1$ results at high pressures. For pressures above $P_c \sim 1.85$ GPa, we observe a rapid decrease in $1/T_1$ without a Hebel-Slichter coherence peak just below $T_c$ (see Fig. 1(b) and Supplementary Fig. 5). At low temperatures, the decrease of $1/T_1$ is very steep, even faster than the $T^5$-variation (see Fig. 1(d) and Supplementary Fig. 6), indicating a full gap. Such behavior is similar to the strong coupling superconductor Ca$_3$Ir$_4$Sn$_{13}$[55] and iron-based superconductor Ba$_{0.68}$K$_{0.32}$Fe$_2$As$_2$[56], where the Hebel-Slichter peak is absent in the fully gapped superconducting state. Nevertheless, the simulations of $1/T_1$ indicate that the gap size of CsV$_3$Sb$_5$ ($2\Delta \sim 3.3 k_B T_c$) is significantly smaller than that of Ca$_3$Ir$_4$Sn$_{13}$ ($2\Delta = 4.42 k_B T_c$). Hence, it is unlikely that the absence of the Hebel-Slichter coherence peak of CsV$_3$Sb$_5$ is attributed to the strong coupling. Instead, it might be similar to that of iron-based superconductors, which could be related to the sign-reversed gap structure of the superconducting gap[57] and the strong spin fluctuations[58], and this is in line with our observation of the enhanced spin fluctuations above $P_c$ in CsV$_3$Sb$_5$(see Fig. 1(a)). To further explore the structure of the superconducting gap, we measured the in-plane upper critical magnetic field $H_{c2}$ as shown below.

## Two-fold symmetry of $H_{c2}$

By rotating the sample with the magnetic field applied in the $ab$ plane, we measured the angular dependence of $H_{c2}$ under pressures at $T \sim 1.6$ K (see Supplementary Fig. 7 for the field dependence of the ac susceptibility at various pressures[47]). Here, $\phi$ represents the angle between the in-plane field orientation and the $a$-axis, as illustrated in Fig. 2(a). The angular dependence of $H_{c2}$ at different pressures, shown in Fig. 2(b) and (c), reveals a clear two-fold symmetry at all pressures. However, the symmetry of the $H_{c2}$ changes across $P_c$. At $P \leq 1.85$ GPa, the maximum $H_{c2}$ value occurs when field direction is nearly perpendicular to the $a$-axis, indicating the maximum superconducting gap alignment in this direction. By contrast, when the pressure is larger than 1.85 GPa, the field direction corresponding to the maximum $H_{c2}$ becomes approximately perpendicular to that at lower pressures, as shown in the Fig. 2(b) and (c). Namely the $H_{c2}$ symmetry changes by 90°.

When the applied pressure is smaller than $P_c$, the two-fold symmetry of superconductivity can be linked to the presence of the CDW order. Firstly, the nematicity in the CDW state can induce an anisotropic Fermi surface[27,35], and then leads to a two-fold symmetry of the $H_{c2}$. Secondly, superconductivity emerges in the coexist state of the

CDW order and loop current state, which can also have a two-fold symmetry of $H_{c2}$[59]. Besides these, the two-fold anisotropy in $H_{c2}$ could also stem from the nodal superconducting gap component indicated by the $T^3$ behavior. We note that the phase of the $H_{c2}$ symmetry reported from the $c$-axis resistivity[35] and specific heat[60] measurements differs from each other and also from our results. This suggests that the $H_{c2}$ symmetry for $P < P_c$ is not due to intrinsic pairing symmetry, but is attributable to the crystal distortion or disorders in the sample. The crystal distortion or disorders can affect the electronic nematicity or loop current order, and thus affect the phase of the $H_{c2}$ symmetry. Obviously, more detailed experimental investigations are still required to disclose the physical nature of the two-fold symmetry of $H_{c2}$ for $P < P_c$.

However, with the increasing pressure above $P_c$, where the CDW order is completely suppressed and CDW fluctuations weaken, we observe a two-fold $H_{c2}$ in which the direction of the maximum $H_{c2}$ changed by nearly 90° (see Fig. 2(c)). Such an observation at high pressures is unpredicted. It is important to note that the observed two-fold symmetry of superconductivity in the $P > P_c$ region cannot be ascribed to the CDW order. Moreover, no other rotational-symmetry breaking state has been reported in the normal state for $P > P_c$, so it is also difficult to attribute the two-fold symmetry in $H_{c2}$ to a Fermi surface anisotropy. Therefore, the two-fold symmetry of $H_{c2}$ is related to the intrinsic superconducting pairing symmetry. In the carrier-doped $Bi_2Se_3$ superconductors[61–64], the two-fold symmetry is due to a pinning of the $d$-vector of the spin-triplet superconductivity[61,65]. However, there is no evidence of spin-triplet in $CsV_3Sb_5$. Considering that the superconducting gap is fully opened and the time-reversal symmetry was found to be broken by high-pressure $\mu$SR measurements[46], we propose a scenario in which a two-component superconductivity of $s + id$ pairing appears for $P > P_c$. Such state was previously proposed in non-trivial multiband superconductors such as iron-based superconductors[66,67] and heavy-fermion superconductors[68,69]. This pairing state violate the time-reversal symmetry and also the $C_4$ rotation symmetry[66]. Although the $s$-wave component might have six-fold symmetry due to the $D_{6h}$ point group symmetry of $CsV_3Sb_5$ as observed by specific heat measurements[60], the $d$-wave component of the $s + id$ state will result in a two-fold symmetry of $H_{c2}$. Meanwhile, the fully-opened superconducting gap of $s + id$ pairing naturally explains the temperature dependence of $1/T_1$ below $T_c$.

**Electric field gradient anomaly**

Figure 3(a) shows the temperature dependence of the NQR frequency at various pressures. At ambient pressure, a reduction in the frequency is observed below $T_c$ as presented in Fig. 3(a). A similar anomaly in the temperature dependence of NQR frequency was also observed in $YBa_2Cu_4O_8$[70], and it is attributed to lattice anomalies resulting from the strong electron-phonon coupling of the electrons. Specifically, the thermal expansion coefficient in $CsV_3Sb_5$ significantly increases along the $c$-axis below $T_c$[71], indicating the further $c$-axis lattice distortions in the superconducting state.

At $P > P_c$ region, instead of the decrease of the NQR frequency, an increase of the NQR frequency is observed below $T_c$. For $P = 2.14$ and 2.31 GPa, the NQR line initially shifts to the lower frequency below $T_c$, but begins to increase below $T \sim 0.9\,T_c$, and eventually a significant enhancement can be seen at $P > 2.5$ GPa. For $P > P_c$ region, the opposite shift of the NQR line compared to the ambient pressure suggests that the main factors causing the observed shift of the NQR lines are different. Due to a more substantial shift in $^{121/123}$Sb-NQR lines in the superconducting state[47], we can obtain the precise temperature dependence of $\nu_Q$ and $\eta$ for $P = 2.87$ GPa, as shown in Fig. 4. Below $T_c$, not only $\nu_Q$, but also $\eta$ increases. The linewidth also increases as the temperature decreases (see Fig. 3(b)), indicating the emergence of the distribution of both $\nu_Q$ and $\eta$. One possible explanation for all these

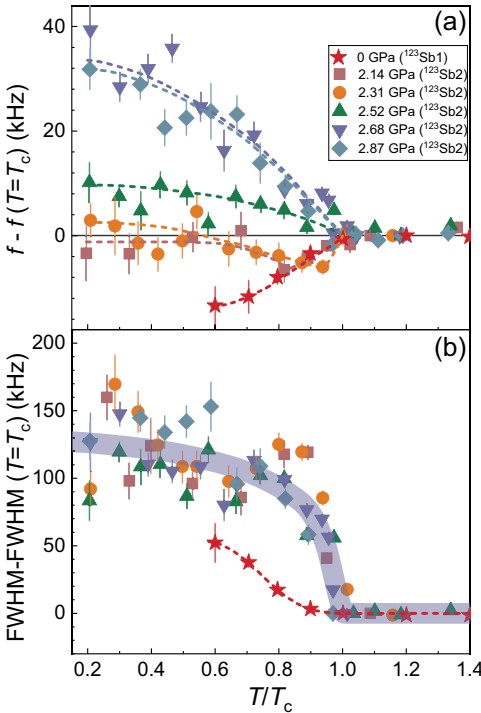

**Fig. 3 | NQR frequency and linewidth in the superconducting state. a** The temperature dependence of $^{123}$Sb1 (ambient pressure) and $^{123}$Sb2 ($P \geq 2.14$ GPa) NQR frequency $f$ after subtracting the $f(T = T_c)$[47]. **b** The temperature dependence of the full width at half maximum (FWHM) of $^{123}$Sb1 (ambient pressure) and $^{123}$Sb2 ($P \geq 2.14$ GPa) NQR spectra after subtracting FWHM$(T = T_c)$ under different pressures. The solid and dashed lines are guides for the eyes. Error bars are s.d. in the fits of the NQR spectra.

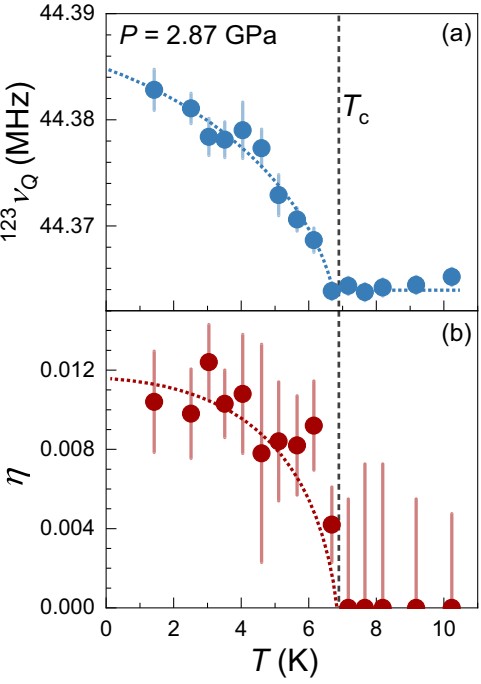

**Fig. 4 | The obtained $\nu_Q$ and $\eta$ in the superconducting state. a, b** are respectively the temperature dependence of $\nu_Q$ and asymmetry parameter $\eta$ of $^{123}$Sb2 at $P = 2.87$ GPa, which are deduced from the $^{121}$Sb2 and $^{123}$Sb2 NQR spectra[47]. The dashed line indicates $T_c$. Error bars are s.d. in the fits of the NQR spectra.

behaviors might be a charge redistribution resulting from the composite pairing in the superconducting state, as predicted in some heavy fermion superconductors by previous theoretical works[72–74]. In materials with mixed valence, it has been proposed that the composite pairing emerges as a low-energy consequence of valence fluctuations in two distinct symmetry channels. This kind of pairing can lead to a mixing of empty and doubly occupied states, causing a redistribution of charge associated with the superconducting transition.

As a result, this charge redistribution is anticipated to induce changes in the EFG around nuclear sites and shifts in NQR frequency below $T_c$. However, there is no evidence indicating that $CsV_3Sb_5$ is a mixed-valent metal. We also note that localized magnetic moments were suggested to play a direct role in pairing involving the condensation of bound states with conducting electrons in heavy fermion superconductors. Although we observed the enhancement of spin fluctuations under high pressures, there is no evidence of the formation of local moments in $CsV_3Sb_5$. Therefore, more works, both theoretical and experimental, are needed in this aspect.

Another possible explanation is more directly associated with the $s + id$ pairing state. A previous theoretical study has shown that the local $C_4$ lattice rotation symmetry would be broken due to the spontaneous currents near disorders in $s + id$ superconducting states[66]. This should lead to an increase of $\eta$ as we observed in Fig. 4(b). In this scenario[66], disorders in the sample would naturally lead to a distribution of both $\eta$ and $\nu_Q$, and result in the quadrupole broadening of the NQR lines in the superconducting state as we found (Fig. 3(b))[47].

## Discussions

We move to discuss the properties and pairing symmetry change across $P_c$. For $P < P_c$, we observed that the $1/T_1$ follows a $T^3$ behavior below $T < 0.4\ T_c$ at ambient pressure, which indicates the line nodes in the superconducting gap. Nevertheless, both the relative weight and the gap size of this nodal gap are found to be significantly smaller than the fully opened gap. Recent theoretical study proposed that there can be possible accidental nodes along the $c$-axis for $P < P_c$ due to the spin fluctuations[75]. For this type of pairing, the existence of nodes within the superconducting gap will not affect the symmetry of $H_{c2}$, which is in accordance with the observation of different phase of the $H_{c2}$ symmetry in different studies[35,60].

For $P > P_c$, we note that the spin fluctuations are prominently enhanced after the CDW is suppressed (see the phase diagram in Fig. 1(a)), which was not taken into account in previous theoretical studies. Generally, spin fluctuations can help form the $d$-wave pairing as observed in cuprates[49]. At ambient pressure where spin fluctuations are weak, the size of the nodal gap is small and its contribution to superconductivity is also relatively small. With the increasing pressure, the enhancement of spin fluctuations leads to an enhancement of $d$-wave. This may cause the $s$-wave and $d$-wave gaps to become degenerate, then would facilitate the $s + id$ pairing. Meanwhile, the enhancement of the $d$-wave component also leads to the reduction of the Hebel-Slichter peak as we observed (see Fig. 1(b)). The $s + id$ state is less well studied. Our findings can inspire more microscopic experimental and theoretical studies.

## Summary

In conclusion, we performed low-temperature and high-pressure $^{121/123}Sb$-NQR measurements to investigate the superconducting pairing symmetry in $CsV_3Sb_5$. At ambient pressure, we observed a distinct kink in $1/T_1$ at $T \sim 0.4\ T_c$, which then follows a $T^3$ behavior down to the lowest temperatures. This behavior indicates multiple superconducting gaps, with line nodes present in the smaller one. When the pressure exceeds $P_c \sim 1.85\ GPa$, where the CDW phase is completely suppressed, $1/T_1$ decreases faster than a $T^5$ dependence below $T_c$. Although no Hebel-Slichter peak was observed, our data indicate that the superconducting gap fully opens for pressures above $P_c$. Most remarkably, the angular

dependence of the in-plane upper critical magnetic field $H_{c2}$ shows an unexpected two-fold symmetry in this high-pressure and fully-gapped superconducting phase. We propose the multi-component $s + id$ pairing to coherently explain the results. An increase in both the NQR frequency and linewidth also occurs in the $P > P_c$ region, which may also be understood as arising from a breaking of the local $C_6$ lattice rotational symmetry due to the $s + id$ pairing. Our results indicate that the $CsV_3Sb_5$ is an unconventional superconductor and its superconducting state is even more exotic at $P > P_c$, which provides new insights into the unconventional superconducting properties of kagome metals.

## Methods

### Sample preparation and NQR measurement

High-quality single crystals of $CsV_3Sb_5$ were synthesized through the self-flux method[76]. The size of the single crystal used for $H_{c2}$ measurements is approximately $2\ mm \times 2\ mm \times 0.1\ mm$. A commercial BeCu/NiCrAl clamp cell from Beijing Easymaterials Technology Co.,Ltd was employed as the pressure cell, and Daphne oil 7373 was utilized as a transmitting medium[77]. The $CsV_3Sb_5$ single crystal flakes were placed inside the pressure cell along with $Cu_2O$ powder, which is used for pressure calibration[78] via its $^{63}\nu_Q$ NQR frequency (refer to Supplementary Fig. 1 for the $^{63}Cu$ NQR spectra of $Cu_2O$ under different pressures)[47]. The solidification of the pressure medium Daphne 7373 occurs at $P > 2.2\ GPa$ at room temperature[77]. When pressurizing the pressure cell, we heat it up to at least 320 K to prevent the pressure medium Daphne 7373 from solidifying, ensuring a hydrostatic environment. NQR measurements were conducted using a phase-coherent pulsed NQR spectrometer. $^{121/123}Sb$ spectra were obtained by sweeping the frequency point by point and integrating the spin-echo signal. The $1/T_1$ was determined using the saturation-recovery method. At ambient pressure, measurements below $T = 1.5\ K$ were conducted using a $^3He$-$^4He$ dilution refrigerator. To test the heat-up effect induced by RF pulses in the superconducting state, we adopt the same method as that of Pustogow et al[79]. During the experiment in the superconducting state, we utilized less than one-third of the highest available energy we found. Further details can be found in Supplementary Fig. 3[47].

### $H_{c2}$ measurements

The single crystal used for the ac susceptibility measurements has naturally formed edges with the angle of about 120° for neighboured edges[35], which allows us to determine the crystallographic axes. The pressure cell was mounted onto the probe of a large bore cryostat such that the $ab$ plane of the crystal was parallel to the magnetic field. The ac susceptibility was measured by the inductance of an $in\ situ$ NQR coil. For the ac susceptibility measurement, $Cu_2O$ is not used for pressure calibration to preclude the influence on the measurement. The applied pressure was obtained by the value of Sb2 NQR frequency at $T = 100\ K$[31]. Angle-dependent measurements were carried out by rotating the pressure cell with a custom-made rotator. The accuracy in the in-plane angle $\phi$ is around a few degrees. The $H_{c2}$ was extracted from the measured ac susceptibility, which is defined as a point off the straight line drawn from high-field value (the normal state)[47]. One may argue that the two-fold symmetry at high pressures can be induced by the misalignment of the field direction to the $c$-axis. Indeed, although we cannot avoid this misalignment, however, this is unlikely for our results. All measurements under various pressures were conducted on the same single crystal through a continuous sequence of pressurizing and depressurizing cycles, without any reassembly of the pressure cell. The misalignment of the sample can not explain the nearly 90° change of the direction of the maximum $H_{c2}$ across $P_c$.

## Data availability

Any additional data that support the findings of this study are available from the corresponding author upon request. Source data are provided with this paper.

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

## Acknowledgements

We thank Y. Yamakawa, S. Onari, H. Kontani and Z. Q. Wang for helpful discussions. This work was supported by the National Key Research and Development Projects of China (Grant No. 2024YFA1611302, No. 2023YFA1406103, No. 2024YFA1409200, No. 2022YFA1403402 and 2022YFA1204100), the National Natural Science Foundation of China (Grant No. 12374142, No. 12304170 and No. 62488201), the Strategic Priority Research Program of the Chinese Academy of Sciences (Grant No. XDB33010100 and No. XDB33030100), Beijing National Laboratory for Condensed Matter Physics (Grant No. 2024BNLCMPKF005) and CAS PIFI program (2024PG0003). This work was supported by the Synergetic Extreme Condition User Facility (SECUF, https://cstr.cn/31123. 02.SECUF).

## Author contributions

The single crystals were grown by Z.Z., H.T.Y. and H.J.G.; The ac susceptibility measurements were performed by X.Y.F. and R.Z.; The NQR measurements were performed by X.Y.F., J.L., Y.Z.Z., J.Y., A.F.F. and R.Z.; R.Z. and G.Q.Z. wrote the manuscript with inputs from X.Y.F.; All authors have discussed the results and the interpretation.

## Competing interests

The authors declare no competing interests.
