## [Transparent Peer Review file · Nature Communications]

Fully-gapped superconductivity with rotational symmetry breaking in pressurized kagome metal CsV_3Sb_5

Corresponding Author: Professor Rui Zhou

Version 0:

Reviewer comments:

Reviewer #1

(Remarks to the Author)

In this paper, by performing low-temperature (T), high-pressure $^{121/123}\text{Sb}$ nuclear quadrupole resonance (NQR) measurements on kagome superconductor CsV_3Sb_5 , the authors found that the spin-lattice relaxation rate $1/T_1$ follows a T^3 variation at low temperatures in the superconducting (SC) state at ambient pressure, which implies a small line node superconducting gap. Even though several NQR measurements have been done before, this node gap has not been observed in previous studies, which could be due to the fact that the temperatures in previous studies are not low enough. This finding on the SC gap structure not only confirms that the main gap is s-wave, which is consistent with the previous reports, but also provides new information about the SC state, and may help understand the SC mechanism in this compound.

Under pressures above beyond $P_c \sim 1.85$ GPa, where the charge-density wave (CDW) phase is suppressed, the Hebel–Slichter coherent peak was not observed, which is also consistent with previous study. By measuring the $1/T_1$ in detail and down to $\sim 0.2 T_c$, the authors found a full gap behavior. Unexpectedly, their angular dependence of the in-plane upper critical magnetic field H_{c2} shows two-fold symmetry. To explain the unusual SC behaviors for pressures beyond P_c , they propose an s + id pairing state.

Other than the superconducting properties, the authors also observed an increase in both the NQR frequency and linewidth below T_c in the $P > P_c$ region, which they infer to a breaking of the local C_6 lattice rotational symmetry due to the s + id pairing.

All these findings are important to understand the superconductivity in CsV_3Sb_5 . Even though the possibility of other pairing states cannot be totally ruled out in this study. The s + id pairing can explain their experimental data, which is one of the possibilities. In general, this paper is well written and the science is clearly presented. I would recommend the publication of this paper after following comments being addressed.

The $1/T_1 T$ data at low temperatures shown in Fig. 1 (c) was reproduced well by the simulation, however, the simulated Hebel–Slichter coherent peak height is much higher than the experimental data. I think the authors need to introduce gap anisotropy or phonon damping to reproduce the height of the HS peak. The readers out of nuclear magnetic resonance field would not think the present simulation is convincing or reliable because there is a big difference between the simulation and the experimental data.

The authors wrote that “Given that the gap size of the nodal gap is extremely small, the signature of this gap can only be observed at very low temperatures, which is indeed in agreement with the observation of a residual DOS in scanning tunneling spectroscopy [37, 38]. This is precisely why it was challenging to detect previously.” I think the signature of this line-node gap can only be observed at very low temperatures is not only due to the extremely small gap size, but also a very small weight of this gap (only 10% of the DOS). If the weight of this small gap is larger, the signature of the small gap can be seen at higher temperatures.

About the superconducting gap structures, the authors fitted the experimental data with two different gaps at ambient pressure, 90% full gap (s-wave) and 10% line-node gap (d-wave, even though it was not clearly written, but strongly implied by the authors, based on my understanding). For $P > P_c$ region, they use s + id pairing. It would be better if the authors could

provide further explanations why the Hebel–Slichter coherent peak can still be observed for two gaps (one s-wave, one d-wave), but not for the s + id pairing.

In supplementary Figure 4, some kinks appear in the magnetic field dependence of the ac susceptibility. What could be the possible reason for this? Since the two-fold symmetry in-plane upper critical field is one of the most important findings in this study, and these features are closely related to the H_{c2} . More discussion/explanation should be provided.

Reviewer #2

(Remarks to the Author)

Referee Report - NCOMMS-25-01443 - Feng et al. - Rotational symmetry breaking within the fully-gapped superconducting state in kagome metal CsV_3Sb_5

Summary:

The authors Feng et al. present a detailed nuclear quadrupole resonance (NQR) and upper-critical field (H_{c2}) ac susceptibility study of the kagome-lattice superconductor CsV_3Sb_3 as a function of pressure and temperature. The authors interpret their data to be consistent with two superconducting gaps (one with line nodes) at ambient pressure, which transition to a single fully opened gap for $P > P_c$. The authors propose an s + id pairing symmetry based on the two-fold symmetry of H_{c2} , and fit the observed changes in the electric field gradient tensor into this story.

Recommendation:

Both the NQR and ac chi data seem to be carefully collected, novel, and clearly presented and do not seem to be significantly overinterpreted. The work is clearly of interest to the community, as it provides further understanding of unconventional superconductivity, also in proximity to magnetic fluctuations and a charge-density-wave state. I recommend the authors take the following comments into account, and publication should be feasible thereafter.

Comments/Questions:

The Authors note that the orientation of the two-fold-symmetric structure in their H_{c2} data does not agree with two previous reports (resistivity and specific heat) in the literature and that disorder is a likely explanation for such disagreement. While the Authors do note that s+id SC would explain their data, it would be misleading to not state this disagreement/complication earlier in the manuscript.

None of the typos or grammatical errors prevent the reader from grasping the Authors' meaning, but should of course be fixed in copyedit.

The reader may be more well informed by a direct comparison between the relaxation data that display a weak Hebel-Slichter peak at ambient pressure and those data taken at higher pressures, to show how the coherence peak evolves with pressure, along the lines of their comparison of the ambient-pressure relaxation data to that of superconducting aluminum. The suppression of the (already weak) coherence peak is critical to the Authors' argument that the nature of the superconducting gap changes as a function of pressure, so a visual representation of such will strengthen their argument.

The data in Fig 1d make it difficult to see the HS peak and the data blend together, please reconsider the best way to present the data/analysis in this subfigure.

Have the Authors checked that their NQR pulses do not result in local heating of the sample, especially in the dil fridge measurements? Could note this in the SI.

The 0.04 mT internal field from muSR could be significantly different at the muon site and the Sb site(s). This should be commented upon.

The authors should include the raw spectra or at least examples thereof in the SI, to help the reader understand their extraction of vQ and η , as these parameters are important for the second argument about charge redistribution in the main text. I only see single resonance lines for all examples shown.

Thank you for the interesting read, and looking forward to hearing your response.

Cheers,
Adam Dioguardi

Version 1:

Reviewer comments:

Reviewer #1

(Remarks to the Author)

I appreciate the authors' effort in revising their manuscript based on the reviewers' comments. Their replies and revised manuscript are satisfactory to me. I am happy to recommend this paper for publication in Nature Communications.

Reviewer #2

(Remarks to the Author)

Thank you for your detailed and careful response. All of my concerns but one have been addressed.

I apologize for previously misunderstanding the Authors' analysis method for extracting ν_Q and η , in which I now gather that they must have performed a global fit to the individual $2\nu_Q$ resonances of the two different Sb isotopes at the Sb(2) site (and thus the same EFG tensor). The known values of the quadrupole moments of the two Sb isotopes provide enough degrees of freedom to extract both ν_Q and η from single resonances of each isotope.

I can now endorse this manuscript for publication in Nature Communications.

Cheers,

Adam Dioguardi

Reply to Reviewers' comments:

First of all, we would like to thank all reviewers for their valuable comments and insightful suggestions. We have taken all the comments/suggestions into account and revised the manuscript accordingly. Specifically,

- 1) As suggested by Reviewer #1, we have re-simulated the temperature-dependent $1/T_1$ in the superconducting state by increasing the width of the breadth function used for $1/T_1$ simulations.
- 2) We have included the temperature-dependent $1/T_1 T$ data at 1.89 GPa in Fig. 1(b) to illustrate the evolution of the coherence peak with pressure.
- 3) We have provided additional details regarding the tests performed to evaluate the heating effect of RF pulses in the superconducting state (see Supplementary Figure 3).
- 4) We have added new figures in the Supplementary Materials to better illustrate the temperature dependence of $1/T_1$, as well as additional Sb-NQR spectra.
- 5) We have added more discussions corresponding to the comments/suggestions.

Below are one-to-one replies to the reviewers' questions and comments.

Reviewer #1 (Remarks to the Author):

In this paper, by performing low-temperature (T), high-pressure 121/123Sb nuclear quadrupole resonance (NQR) measurements on kagome superconductor CsV3Sb5, the authors found that the spin-lattice relaxation rate $1/T_1$ follows a T^3 variation at low temperatures in the superconducting (SC) state at ambient pressure, which implies a small line node superconducting gap. Even though several NQR measurements have been done before, this node gap has not been observed in previous studies, which could be due to the fact that the temperatures in previous studies are not low enough. This finding on the SC gap structure not only confirms that the main gap is s-wave, which is consistent with the previous reports, but also provides new information about the SC state, and may help understand the SC mechanism in this compound.

Under pressures above beyond $P_c \sim 1.85$ GPa, where the charge-density wave (CDW) phase is suppressed, the Hebel–Slichter coherent peak was not observed, which is also consistent with previous study. By measuring the $1/T_1$ in detail and down to $\sim 0.2 T_c$, the authors found a full gap behavior. Unexpectedly, their angular dependence of the in-plane upper critical magnetic field H_{c2} shows two-fold symmetry. To explain the unusual SC behaviors for pressures beyond P_c , they propose an s + id pairing state.

Other than the superconducting properties, the authors also observed an increase in both the NQR frequency and linewidth below T_c in the $P > P_c$ region, which they infer to a breaking of the local C_6 lattice rotational symmetry due to the s + id pairing.

All these findings are important to understand the superconductivity in CsV₃Sb₅. Even though the possibility of other pairing states cannot be totally ruled out in this study. The $s + id$ pairing can explain their experimental data, which is one of the possibilities. In general, this paper is well written and the science is clearly presented. I would recommend the publication of this paper after following comments being addressed.

Reply:

Thank you for devoting your time to reading and reviewing the manuscript, and for offering valuable suggestions and comments.

The $1/T_1 T$ data at low temperatures shown in Fig. 1 (c) was reproduced well by the simulation, however, the simulated Hebel–Slichter coherent peak height is much higher than the experimental data. I think the authors need to introduce gap anisotropy or phonon damping to reproduce the height of the HS peak. The readers out of nuclear magnetic resonance field would not think the present simulation is convincing or reliable because there is a big difference between the simulation and the experimental data.

Reply:

Thank you for the insightful suggestion regarding the $1/T_1$ simulations. To account for a real system, anisotropies in the energy gap due to imperfections in the crystal structure as well as finite lifetimes of the quasiparticles need to be taken into consideration. Both of these factors cause a broadening of the singularity peaks in the density of states and thereby a reduction in the height of the HS coherence peak. In our simulation, we convolute the density of states with a breadth function proposed by Hebel and Slichter (PRB 116, 79 (1959).), which is a rectangular one of width $2\delta E$ and height $1/2\delta E$. In the revised manuscript, we have re-simulated the temperature dependence of $1/T_1$ for all pressures with a larger width of the breadth function used for $1/T_1$ simulations. The updated results show improved agreement with the experimental HS peak height (see Fig. 1(c) and (d)).

The authors wrote that “Given that the gap size of the nodal gap is extremely small, the signature of this gap can only be observed at very low temperatures, which is indeed in agreement with the observation of a residual DOS in scanning tunneling spectroscopy [37, 38]. This is precisely why it was challenging to detect previously.” I think the signature of this line-node gap can only be observed at very low temperatures is not only due to the extremely small gap size, but also a very small weight of this gap (only 10% of the DOS). If the weight of this small gap is larger, the signature of the small gap can be seen at higher temperatures.

Reply:

We thank the reviewer for highlighting this important point. The small weight of

this line-node gap is indeed a crucial factor that makes it more difficult to observe at higher temperatures. In the revised manuscript, we have clarified this aspect as follows: *“Given that both the size and the weight of the nodal gap are extremely small compared to the fully opened gap, the signature of this gap can only be observed at very low temperatures, which is indeed in agreement with the observation of a residual DOS in scanning tunneling spectroscopy[37, 38]. This is precisely why it was challenging to detect previously.”*

About the superconducting gap structures, the authors fitted the experimental data with two different gaps at ambient pressure, 90% full gap (s-wave) and 10% line-node gap (d-wave, even though it was not clearly written, but strongly implied by the authors, based on my understanding). For $P > P_c$ region, they use s + id pairing. It would be better if the authors could provide further explanations why the Hebel–Slichter coherent peak can still be observed for two gaps (one s-wave, one d-wave), but not for the s + id pairing.

Reply:

We apologize for not having made further explanation. Under ambient pressure, while a small superconducting gap with line nodes exists, the relative weight and gap size of the fully-opened gap are significantly larger than those of the nodal gap. Additionally, theoretical calculations on the kagome lattice suggest the possibility of unconventional pairing states with a sign-changing gap structure, particularly a *d*-wave gap. This could manifest as a small Hebel-Slichter peak in the temperature dependence of $1/T_1$ due to destructive sublattice interference effects (see Ref. 54). These factors likely contribute to the small coherence peak observed in $1/T_1$.

In the $P > P_c$ region, the absence of the HS coherence peak in the fully-gapped state resembles observations in the strong-coupling superconductor $\text{Ca}_3\text{Ir}_4\text{Sn}_{13}$ (see Ref. 55) and iron-based superconductors (see Ref. 56). Given that the gap size of CsV_3Sb_5 ($2\Delta \sim 3.3 k_B T_c$) is considerably smaller than that of $\text{Ca}_3\text{Ir}_4\text{Sn}_{13}$ ($2\Delta = 4.42 k_B T_c$), it is unlikely that the absence of the Hebel-Slichter coherence peak in CsV_3Sb_5 is due to strong coupling. Instead, it may be similar to iron-based superconductors, where the absence of the coherence peak is likely linked to the sign-reversed gap structure of the superconducting gap (see Ref. 57) and strong spin fluctuations (see Ref. 58). Indeed, we observed that the spin fluctuations are prominently enhanced after the CDW is suppressed for $P > P_c$. The enhancement of spin fluctuations might also lead to an enhancement of *d*-wave. All these lead to the reduction of the HS coherence peak in the $P > P_c$ region. In the revised manuscript, we have added the following discussion as follows: *“Such behavior is similar to the strong coupling superconductor $\text{Ca}_3\text{Ir}_4\text{Sn}_{13}$ [55] and iron based superconductor $\text{Ba}_{0.68}\text{K}_{0.32}\text{Fe}_2\text{As}_2$ [56], where the Hebel-Slichter peak is absent in the fully gapped superconducting state. Nevertheless, the simulations of $1/T_1$ indicate that the gap size of CsV_3Sb_5 ($2\Delta \sim 3.3 k_B T_c$) is significantly smaller than that of $\text{Ca}_3\text{Ir}_4\text{Sn}_{13}$ ($2\Delta = 4.42 k_B T_c$). Hence, it is unlikely that the absence of the Hebel-Slichter coherence peak of CsV_3Sb_5 is attributed to the strong coupling. Instead, it might be similar to that of iron-based superconductors, which could be*

related to the sign-reversed gap structure of the superconducting gap[57] and the strong spin fluctuations[58], and this is in line with our observation of the enhanced spin fluctuations above P_c in CsV_3Sb_5 (see Fig. 1(a)). To further explore the structure of the superconducting gap, we measured the in-plane upper critical magnetic field H_{c2} as shown below.” and “Meanwhile, the enhancement of d-wave component also leads to the reduction of the Hebel-Slichter peak as we observed (see Fig. 1 (b)).”

In supplementary Figure 4, some kinks appear in the magnetic field dependence of the ac susceptibility. What could be the possible reason for this? Since the two-fold symmetry in-plane upper critical field is one of the most important findings in this study, and these features are closely related to the H_{c2} . More discussion/explanation should be provided.

Reply:

We thank the reviewer for the insightful suggestion. Similar kinks have also been observed in the magnetic field dependence of the ac susceptibility in Sr_2RuO_4 , which are due to vortex motion (PRB 107, 064509, 2023). Since these kinks occur under magnetic fields much lower than H_{c2} , they do not influence the determination of H_{c2} . As pressure varies, it is possible that the phase diagram of the vortex structure undergoes changes. We plan to investigate this aspect further in future research. In the revised manuscript, we have included the following discussion to address this point: *“For small fields, a small increase of the diamagnetic response is visible for some pressures and forms kinks in the field dependence of the ac susceptibility. This might be associated with vortex motion, which was also observed in Sr_2RuO_4 [3]. Since these occur under a magnetic field much lower than H_{c2} , they will not affect the determination of the H_{c2} .”*

Reviewer #2 (Remarks to the Author):

Referee Report - NCOMMS-25-01443 - Feng et al. - Rotational symmetry breaking within the fully-gapped superconducting state in kagome metal CsV_3Sb_5

Summary:

The authors Feng et al. present a detailed nuclear quadrupole resonance (NQR) and upper-critical field (H_{c2}) ac susceptibility study of the kagome-lattice superconductor CsV_3Sb_5 as a function of pressure and temperature. The authors interpret their data to be consistent with two superconducting gaps (one with line nodes) at ambient pressure, which transition to a single fully opened gap for $P > P_c$. The authors propose an $s + id$ pairing symmetry based on the two-fold symmetry of H_{c2} , and fit the observed changes in the electric field gradient tensor into this story.

Recommendation:

Both the NQR and ac chi data seem to be carefully collected, novel, and clearly

presented and do not seem to be significantly overinterpreted. The work is clearly of interest to the community, as it provides further understanding of unconventional superconductivity, also in proximity to magnetic fluctuations and a charge-density-wave state. I recommend the authors take the following comments into account, and publication should be feasible thereafter.

Comments/Questions:

The Authors note that the orientation of the two-fold-symmetric structure in their H_{c2} data does not agree with two previous reports (resistivity and specific heat) in the literature and that disorder is a likely explanation for such disagreement. While the Authors do note that $s+id$ SC would explain their data, it would be misleading to not state this disagreement/complication earlier in the manuscript.

Reply:

We apologize for any confusion caused by the previous explanation. We note that the phase of the H_{c2} symmetry reported from c -axis resistivity measurements (see Ref. [35]) and specific heat measurements (see Ref. [60]) differs not only from each other but also from our results in the $P < P_c$ region. This indicates that the H_{c2} symmetry for $P < P_c$ is unlikely to be associated with intrinsic pairing symmetry, such as the $s + id$ superconducting state. Instead, crystal distortions or disorders may influence the electronic nematicity or loop current order, thereby affecting the phase of the H_{c2} symmetry. Consequently, we propose that the H_{c2} symmetry in the $P < P_c$ region is attributable to crystal distortions or sample disorders.

For the $P > P_c$ region, no other rotational symmetry-breaking states have been reported in the normal state, making it difficult to attribute the two-fold symmetry in H_{c2} to Fermi surface anisotropy. Therefore, the two-fold symmetry of H_{c2} is most likely linked to the intrinsic superconducting pairing symmetry. Considering the fully-opened superconducting gap and the breaking of time-reversal symmetry, we propose that a two-component $s + id$ superconducting state emerges in the $P > P_c$ region.

None of the typos or grammatical errors prevent the reader from grasping the Authors' meaning, but should of course be fixed in copyedit.

Reply:

We apologize for any typos or grammatical errors in the original manuscript. These have been carefully corrected in the revised version.

The reader may be more well informed by a direct comparison between the relaxation data that display a weak Hebel-Slichter peak at ambient pressure and those data taken at higher pressures, to show how the coherence peak evolves with pressure, along the lines of their comparison of the ambient-pressure relaxation data to that of superconducting aluminum. The suppression of the (already weak) coherence peak is critical to the Authors' argument that the nature of the

superconducting gap changes as a function of pressure, so a visual representation of such will strengthen their argument.

Reply:

We thank the reviewer for the valuable suggestion. We have added the representative temperature-dependent $1/T_1T$ data at 1.89 GPa in Fig. 1(b) to compare with the ambient pressure data and illustrate the evolution of the HS coherence peak under pressure. Additionally, we have included a new figure in the Supplementary Materials to present the temperature dependence of $1/T_1T$ for $1.89 \text{ GPa} \leq P \leq 2.68 \text{ GPa}$ (see Supplementary Fig. 5).

The data in Fig 1d make it difficult to see the HS peak and the data blend together, please reconsider the best way to present the data/analysis in this subfigure.

Reply:

We thank the reviewer for the helpful suggestion. We have added a new figure to the Supplementary Materials (see Supplementary Fig. 6), which presents the temperature dependence of $1/T_1$ for each pressure more clearly.

Have the Authors checked that their NQR pulses do not result in local heating of the sample, especially in the dil fridge measurements? Could note this in the SI.

Reply:

We apologize for not including the test of the heat-up effect in the original manuscript. We conducted a heat-up effect test prior to our measurements in the superconducting state, following the method described in Pustogow et al., Nature 574, 72 (2019). For the measurements in the superconducting state, we utilized only one-third of the highest available energy we found. A detailed description of this test has been added to the Supplementary Materials (see Supplementary Figure 3).

The 0.04 mT internal field from μ SR could be significantly different at the muon site and the Sb site(s). This should be commented upon.

Reply:

We thank the reviewer for the valuable suggestion. Indeed, the internal field at the muon site can be significantly different from that at the Sb sites. Given that the error bars in the line width of the Sb sites are about 10 kHz, our measurements are unable to detect an internal field less than 1 mT at the Sb sites. Although the internal field could be significantly different at the muon and the Sb sites, the internal field at the Sb sites might still be extremely small and undetectable in our NQR measurements. We have included a comment on this point in the revised manuscript as follows: "*Given that the error bars in the line width of the Sb site are ~ 10 kHz, our measurements are unable to detect an internal field less than 1 mT at the Sb sites. It is noted that the internal field obtained from the μ SR experiment was merely 0.04*

mT[4]. Although the internal field could be significantly different at the muon and the Sb sites, the internal field at the Sb sites might still be extremely small and undetectable in our NQR measurements.”

The authors should include the raw spectra or at least examples thereof in the SI, to help the reader understand their extraction of ν_Q and η , as these parameters are important for the second argument about charge redistribution in the main text. I only see single resonance lines for all examples shown.

Reply:

We apologize for not including the corresponding NQR spectra in the original manuscript. In the revised manuscript, we have added a new figure to the Supplementary Materials (see Supplementary Figure 11(a)) showing the spectra of $^{121/123}\text{Sb}_2$ at $P = 2.87$ GPa, from which the ν_Q and η were extracted.

Thank you for the interesting read, and looking forward to hearing your response.

Cheers,
Adam Dioguardi

Reply:

Thank you for devoting your time to reading and reviewing the manuscript, and for offering valuable suggestions and comments.

Reply to Reviewers' comments:

First of all, we would like to thank all reviewers for their valuable comments and insightful suggestions. We have taken all the suggestions into account and revised the manuscript accordingly. Specifically,

1) We have added a sentence about how to obtain ν_q and η in Supplementary Figure 11.

Below are one-to-one replies to the reviewers' questions and comments.

Reviewer #1 (Remarks to the Author):

I appreciate the authors' effort in revising their manuscript based on the reviewers' comments. Their replies and revised manuscript are satisfactory to me. I am happy to recommend this paper for publication in Nature Communications.

Reply:

Thank you for devoting your time to reading and reviewing the manuscript.

Reviewer #2 (Remarks to the Author):

Thank you for your detailed and careful response. All of my concerns but one have been addressed.

I apologize for previously misunderstanding the Authors' analysis method for extracting ν_Q and η , in which I now gather that they must have performed a global fit to the individual $2\nu_Q$ resonances of the two different Sb isotopes at the Sb(2) site (and thus the same EFG tensor). The known values of the quadrupole moments of the two Sb isotopes provide enough degrees of freedom to extract both ν_Q and η from single resonances of each isotope.

Reply:

We apologize for any confusion caused in the previous version. To address this point more clearly, we have added the following sentence to Supplementary Figure 11: "*Since the quadrupole moments of the two Sb isotopes $^{121/123}\text{Sb}$ are known, both ν_q and η can be extracted from the $^{121}\text{Sb}2$ and $^{123}\text{Sb}2$ lines as shown in (a).*"

I can now endorse this manuscript for publication in Nature Communications.

Cheers,
Adam Dioguardi

Reply:

Thank you for devoting your time to reading and reviewing the manuscript.